# Attachment-related perceptions of life events

Logan C. Gibson(ID), William J. Chopik(ID)*

Department of Psychology, Michigan State University, East Lansing, Michigan, United States of America

* chopikwi@msu.edu

## Abstract

This study explored how attachment orientations are associated with perceptions of life events. Although there is evidence that life and relationship contexts have the potential to alter attachment anxiety and avoidance across the lifespan, life events often exert only modest or transient effects on attachment orientations. The current study ($N = 1943$; $M_{age} = 19.61$; 74.1% women) examined associations between attachment orientations, perceptions of whether life events might engender personality changes, and perceptions of 20 hypothetical life events across nine dimensions (e.g., emotional significance, impact, control). Individuals high in attachment anxiety perceived life events as more challenging, impactful, emotionally significant, unpredictable, negative, and likely to alter their worldview and negatively affect their social status—viewing them as likely to induce personality changes. Conversely, individuals high in attachment avoidance *minimized* life events' potential effects, perceiving them as less significant and less likely to alter personality. Future research could further examine whether attachment orientations shift in response to life events according to perceived event characteristics.

## Introduction

There has been a great deal of public attention and theorizing about whether people's attachment orientations—their characteristic approach to relationships—change over time and across the lifespan [1–3]. Although there is a general consensus that attachment-related changes are initiated by changes in social and environmental circumstances [4–6], formal examinations of how life events in particular change a person's attachment orientation have found relatively underwhelming support for this idea. In other words, only a few life events are associated with changes in attachment orientations, and when they are, they have only temporary and relatively small effects [7]. However, one promising advancement for trying to understand why life events do or don't change psychological characteristics is the development of taxonomies explaining people's *perceptions* of life events [8,9]. But the degree to which individual differences in attachment are associated with a broad array of perceptions of life events is relatively unexplored. The current study examined attachment-related

**Data availability statement:** All data, syntax, and materials files are available from the Open Science Framework (DOI: 10.17605/OSF.IO/EBG86); also see https://osf.io/ebg86.

**Funding:** The author(s) received no specific funding for this work.

**Competing interests:** The authors have declared that no competing interests exist.

differences in life event perceptions and broader assessments about whether life events would change their personalities.

## The cognitive and motivational tendencies of people high in attachment anxiety and avoidance

An individual's attachment orientation is how they generally approach close relationships and is thought to be formed by early interactions with caregivers, though experiences with relationships later in life likely also affect attachment orientations [1,2,10–15]. Variation in attachment orientations is characterized by how people vary on two dimensions—attachment anxiety and attachment avoidance [16]. Attachment anxiety is characterized by fear of rejection and a preoccupation with seeking reassurance from close others [17]. Attachment avoidance is characterized by a discomfort with closeness and dependence [18]. Since its extension from childhood to adult romantic and platonic relationships, attachment theory has served as an organizing framework for characterizing thoughts, feelings, and behavior in the context of close relationships [19,20]. People low in both attachment anxiety and attachment avoidance are considered to be relatively securely attached.

Although there are normative shifts in attachment orientations across the lifespan, attachment orientations are generally considered stable individual difference characteristics [12,21–23]. One of the reasons why attachment orientations are stable is because of the cognitive and motivational mechanisms that reify one's existing "working model" of attachment to promote stability [4,11,24,25]. In other words, like other psychological characteristics, attachment orientations guide the situations people find themselves in, how they interpret those situations, and can also fundamentally change aspects of those situations [26,27]. People higher in attachment anxiety and avoidance tend to interpret situations in ways that are consistent with what they think about themselves and other people [24,28–31].

For example, people higher in attachment anxiety tend to engage in hyperactivation of the attachment system as a way of interpreting the world and regulating their emotions and behavior. Attachment-related threats typically trigger efforts to increase proximity to attachment figures and to secure support from them [32,33]. Collectively, these strategies are meant to prevent abandonment [34]. People high in attachment anxiety tend to hold negative perceptions of themselves, think their self-worth is conditional, and have heightened rejection sensitivity to what they feel are potentially unavailable attachment figures [35–37]. Hyperactivation of the attachment system is accompanied by a hyperfixation on the availability of close others, intense scrutiny over support provision, selective attention toward rejection-related cues, and constant threat-monitoring [38–41]. Among people higher in attachment anxiety, this hyperactivation is often made manifest in intense feelings of jealousy (a vigilance to the attention and commitment of others), maladaptive attributions of ambiguous behavior (assuming others are inconsistently reliable as a way of monitoring availability), and otherwise perceiving the world in a more negative light, as this is consistent with their internal way of thinking about the world [30,42,43]. Although these attributional and motivational tendencies make sense in light of their attachment histories and working

models, this hyperactivation of the attachment system often results in greater error in threat detection, an ironic *undermining* of relational commitment from partners, rumination and catastrophizing, and an inaccurate warping of relational circumstances—all correlates of less satisfying interpersonal relationships [17,44–47]. As a result, when confronted with either an ambiguous life event or situation, people high in attachment anxiety would likely perceive it to be more negative for them, their relationships, and their future.

In contrast, people higher in attachment avoidance engage in chronic attempts to *deactivate* the attachment system. When attachment needs are triggered, people high in attachment avoidance typically engage in strategies designed to suppress dependency, dampen emotional activation, and maintain distance from attachment figures [18,34,48,49]. Collectively, these strategies are meant to prevent vulnerability and protect autonomy rather than pursue closeness [32,50]. People high in attachment avoidance tend to hold a negative perception of close others—often believing others are unreliable, intrusive, or emotionally overwhelming—while maintaining a compensatory positive self-view centered on independence and self-sufficiency [35,51,52]. Deactivation of the attachment system is accompanied by selective attention away from emotional cues, minimizing the personal relevance of relational events, and the suppression or dismissal of attachment-related thoughts and feelings [53–55]. Among people higher in attachment avoidance, this deactivation often manifests in emotional disengagement, reluctance to self-disclose, downplaying relational difficulties, and attributing ambiguous situations in ways that preserve psychological distance and self-reliance [40,56,57]. Although these tendencies serve an immediate regulatory function by reducing perceived relational threat, they may also result in reduced emotional awareness, impaired intimacy, and difficulty acknowledging support when it is offered—patterns that predict lower relationship satisfaction and reduced partner responsiveness over time [58–61]. As a result, when confronted with an ambiguous event or situation, people high in attachment avoidance would likely perceive it as less personally or relationally relevant and respond with emotional disengagement rather than threat sensitivity.

In these ways, people high in attachment anxiety or avoidance have several cognitive and motivational tendencies that shape how they perceive themselves, others, and the world. And these tendencies—and the perceptions they shape— likely explain why people's attachment orientations are stable, even when disruptive and significant life shocks or events happen to them [62–64], even if some relational contexts might alter these enduring processes [5,60,65]. But the study of life events and their impact on attachment changes has been limited in several ways [8]. As an illustration of just how powerful these mindsets are, Fraley and colleagues [7] followed over 4,000 people over a mean time frame of 2 years to examine how their attachment orientations changed before-to-after 25 different life events that encapsulated both relational and non-relational contexts. Although many life events had an initial association with changes in attachment orientations, very few had *enduring* effects. Importantly, this work rarely examines people's qualitative assessments of life events—a potential avenue through which attachment orientations might exert their influence. In the current study, we examined how people high in attachment anxiety and avoidance perceive life events as an illustrative case to assess their cognitive and motivational tendencies—tendencies that explain sources of anxiety and avoidance and why they might be durable to changes in situations and life events. But comprehensive models for conceptualizing the range of perceptions people have in response to events are rare. However, researchers have recently developed a taxonomy for characterizing people's perceptions of life events.

## Attachment-related differences in life event perceptions

Recent developments in characterizing life events based on the way people perceive them is a potentially useful advancement for explaining how life events affect people [9]. One such taxonomy is contained in Luhmann et al. (2021)'s Event Characteristics Questionnaire (ECQ). For this particular taxonomy, there are nine dimensions on which perceptions of life events vary: valence, challenge, extraordinariness, impact, external control, social status change, emotional significance, predictability, and the capacity to change a person's worldview. Since its inception, the taxonomy has greatly expanded our understanding of individual differences in life event perceptions, how perceptions are related to changes in personality

and well-being, and associations with mental health [66–69]. For example, perceiving an event to be more negatively valenced, more of a challenge, extraordinary, and negatively affecting one's social status were associated with declines in life satisfaction after a life event [9,70].

There have also been links established between the Big Five personality traits and these life event perceptions [71]. Reviewing associations with the Big Five personality traits is valuable given their empirical overlap with attachment orientations and because they share some of the same attributional and evaluative tendencies mentioned above [72]. For example, highly agreeable people—a characteristic shared by those lower attachment avoidance—thought social life events (e.g., divorce, falling out with a friend, marriage, making a new close friend, becoming a parent) were more controllable and less harmful to their social status. Likewise, people high in neuroticism—a characteristic shared by those higher in attachment anxiety—thought life events were more challenging, emotionally significant, impactful, and had negative effects on their social status. Some of these findings are intuitive, but not all personality traits were related to life event perceptions in intuitive ways (e.g., conscientiousness was not particularly related to work or school event perceptions). One reasonable critique of the attachment literature is that attachment-related differences might be reducible to personality-related differences (e.g., the heightened negative perception of people high in attachment anxiety might be merely a reflection of their high levels of neuroticism). Worth noting, associations between attachment orientations and Big Five personality traits are modest, such that they are not empirically redundant. And importantly, attachment orientations provide additional explanatory power for life and relational outcomes over-and-above Big Five personality traits [41,72,73].

To date, however, there has not been an examination of how individuals high in attachment anxiety and avoidance differ in their perceptions of life events using this expanded taxonomy. Nevertheless, there are some reasons why attachment orientations might have implications for how people evaluate hypothetical life events as specified by the Luhmann et al. taxonomy. For example, individuals high in attachment anxiety tend to excessively seek reassurance from others [17]. As a result, individuals high in attachment anxiety might evaluate ostensibly negative relational events (e.g., dissolutions, falling out with a friend) as more negatively valenced, more impactful, and more challenging. Individuals high in attachment avoidance tend to employ deactivating strategies to minimize distress [32,48]. As a result, individuals high in attachment avoidance might evaluate ostensibly distressing life events (e.g., losing a family member or friend), being the victim of a crime) as less emotionally significant and less likely to impact their worldview—as acknowledging the impact of life events might be accompanied with distress. These differences would be consistent with the appraisal and attributional tendencies of individuals high in attachment anxiety and avoidance discussed above. Our study moves beyond evaluating whether life events are more or less likely to happen to individuals high in attachment anxiety and avoidance. Rather, we examined associations between attachment orientations and this emerging classification system to clarify attachment-related differences in qualitative perceptions of hypothetical life events. Doing so situates attachment orientations as a potential predictor of Luhmann et al.'s taxonomy and can encourage future research examining adaptation to these life events among individuals high in attachment anxiety or avoidance.

Indeed, some evidence suggests that individuals high in attachment anxiety and avoidance might differ on some of the perceptual characteristics outlined by Luhmann et al. For example, insecurely attached people tend to make more negative attributions about both themselves and their loved ones [28–30,42,74]. This includes making more internal and negative attributions about their attitudes and feelings toward love and themselves [35], more negative expectations about their relational futures [75], and unfavorable explanations about their partner's behavior [24]. This literature is relevant as attributions of control and unpredictability are two of the characteristics in the Luhmann et al. (2021) model. Insecurely attached adults also tend to have a negativity bias [76] and remember relationship events as more negative than they were when they happened [29,44]—phenomena closely relevant to Luhmann et al. (2021)'s assessment of valence and aligned well with appraisal models of stress [32]. Finally, people high in attachment anxiety seem more affected by interpersonal contexts and partner feedback and perceive relationship contexts to be more threatening, suggesting a possible negative sensibility and sensitivity to life events [77,78]. The opposite can be said for people high in attachment avoidance

who both consciously and unconsciously avoid relationship information and are less reactive to partner feedback [77,79]. These findings are most directly relevant to Luhmann et al. (2021)'s assessments of the perceived impact and challenge of life events.

Thus, altogether, there is some evidence that attachment orientations are associated with how people perceive life events and relational contexts. However, some perceptions of life events from an attachment perspective have not been studied as much (e.g., how they might change a person's worldview), many of these perceptions have not been studied in the context of life events specifically, and it is unclear whether these attachment-related differences in perceptions vary across different life events.

## The current study

Attachment orientations change over time and across the lifespan. Researchers think that at least some of these changes are attributable to life events, both relational and non-relational. However, evidence for life events changing people has been a bit underwhelming to date, possibly because qualitative information about those life events is often missing. One solution is to examine perceptions and characteristics of life events and how they relate to attachment orientations, about which there has been incomplete coverage to date. We examined associations between attachment orientations and perceptions of 20 different life events, both in terms of whether they have the potential to cultivate psychological change and specific characteristics of those life events. We capitalized on the procedure Rakhshani, Lucas [71] developed to examine associations with Big Five personality traits. Specifically, we examined (a) whether attachment orientations were associated with perceptions of personality change (to provide an initial assessment of whether life events would change them at all) and (b) how attachment orientations were associated with Luhmann et al. (2021)'s event characteristics in response to 20 hypothetical events.

## Method

Data, syntax, and materials for the current project can be found at https://osf.io/ebg86. This study was not pre-registered. Thus, all of our research questions were exploratory, and we had no hypotheses. This study was carried out in accordance with the recommendations of Michigan State University's Institutional Review Board (IRB# x16-1291e) and run online with informed consent being secured from all participants virtually/electronically (by clicking a next arrow; documentation requirement waived by the local ethics committee). Data were analyzed anonymously, and all participants were adults (i.e., no minors). Data were collected from March 27, 2023 until April 25, 2025. Portions of these data, but not the attachment data, are also published elsewhere [80].

## Participants and procedure

Participants were 1,943 undergraduate students who participated in the research study in exchange for course credit. They ranged in age from 18 to 35 ($M_{age}$ = 19.61, $SD$ = 1.46) and were mostly women (74.1%) followed by men (24.7%) and non-binary (1.2%). The sample was mostly White (69.1%), followed by Asian (10.6%), Black (7.2%), multi-racial (5.5%), Hispanic/Latinx (4.3%), and 3.3% other races/ethnicities.

Following the procedure by Rakhshani, Lucas [71], participants first filled out a series of individual difference measures (including attachment orientation) and other measures not pertinent to the current report. Similar to Rakhshani et al., participants also reported how likely each of the life events might change their personalities and indicated if they experienced a particular life event themselves (and if so, when). Because of the infrequency of experiencing most of the life events, we chose to focus on perceptions of hypothetical events as it guaranteed a sufficient sample size for each analysis. This information is available in the shared data file on the OSF page.

Most events were not experienced by participants given their infrequency (particularly among college students). We only had more than 100 participants for 10 different life events. We were unable to make a comparison between ratings of

hypothetical and experienced events given the few experienced life events represented (and the disparity in sample sizes between the two types [hypothetical and experienced] of events). Nevertheless, these data are available on the OSF site. Then, they received two events randomly from the broader list (that they did not experience themselves; see Table 1 for a full list) and rated their perceptions of those events based on event characteristics (see below).

### Measures

**Attachment orientations.** Attachment orientations were measured using the 9-item version of Experiences in Close Relationships, a short-form scale that has been shown to adequately predict relationship and interpersonal functioning outcomes [i.e., investment characteristics, depression; 81]. The 3-item anxiety subscale reflects an individual's concern about abandonment (sample item: "I'm afraid that other people may abandon me"). The 6-item avoidance subscale reflects an individual's discomfort with emotional and physical closeness (sample item: "It helps to turn to people in times of need"; reverse-scored). Participants were asked to rate the extent to which they agree with each item on a scale ranging from 1(*strongly disagree*) to 7(*strongly agree*), and items are averaged to create subscales for anxiety ($M = 4.47$, $SD = 1.57$; $\alpha = .89$) and avoidance ($M = 3.51$, $SD = 1.12$; $\alpha = .85$). Attachment anxiety and avoidance were negligibly associated with each other ($r = .13$, $p < .001$).

**Table 1. Life events, perceptions of change, and associations with attachment orientations.**

| Life event | M | SD | Attachment anxiety | | Attachment avoidance | |
|---|---|---|---|---|---|---|
| | | | Raw | Partial | Raw | Partial |
| Married | 0.941 | 1.013 | 0.041 | **0.050** | **−0.058** | **−0.061** |
| Divorced | 1.234 | 0.980 | **0.062** | **0.076** | **−0.103** | **−0.108** |
| Entered the workforce | 0.773 | 0.961 | 0.023 | 0.043 | **−0.100** | **−0.106** |
| Fired from job | 0.904 | 0.985 | **0.055** | **0.074** | **−0.120** | **−0.125** |
| Laid off from job | 0.874 | 1.019 | **0.047** | **0.062** | **−0.104** | **−0.108** |
| Moved to a new city/town at least 50 miles (80 km) away | 0.972 | 0.993 | **0.059** | **0.071** | **−0.099** | **−0.106** |
| Close friend died | 1.462 | 0.908 | **0.056** | **0.068** | **−0.118** | **−0.126** |
| Romantic partner died | 1.590 | 0.887 | 0.039 | **0.055** | **−0.129** | **−0.136** |
| Father or mother died | 1.523 | 0.908 | 0.034 | 0.045 | **−0.115** | **−0.121** |
| Close family member died | 1.190 | 0.948 | 0.036 | **0.051** | **−0.135** | **−0.144** |
| Became a parent | 1.505 | 0.884 | 0.012 | 0.031 | **−0.128** | **−0.131** |
| Became seriously ill or injured | 1.252 | 0.954 | 0.025 | **0.045** | **−0.146** | **−0.150** |
| Jailed or imprisoned | 1.310 | 0.974 | 0.013 | 0.030 | **−0.129** | **−0.130** |
| Close family member jailed or imprisoned | 0.652 | 1.025 | −0.008 | 0.008 | **−0.135** | **−0.138** |
| Spent significant time in a different country | 0.862 | 0.977 | −0.008 | 0.009 | **−0.129** | **−0.130** |
| Made a new close friend | 0.362 | 0.971 | 0.031 | **0.048** | **−0.091** | **−0.098** |
| Had a falling out with a close friend | 0.532 | 0.968 | **0.078** | **0.099** | **−0.091** | **−0.104** |
| Started college or university | 0.996 | 0.904 | **0.080** | **0.093** | **−0.092** | **−0.104** |
| Graduated college or university | 0.689 | 1.014 | 0.042 | **0.056** | **−0.089** | **−0.096** |
| Victim of a serious crime | 1.342 | 0.943 | **0.076** | **0.092** | **−0.131** | **−0.140** |
| Experienced a natural disaster | 0.976 | 0.999 | **0.064** | **0.082** | **−0.142** | **−0.153** |
| Total/Average | 1.045 | 0.963 | 0.041 | 0.057 | −0.114 | −0.120 |

*Note*. *N*s range from 1928–1932. Bolded values are significant at *p* < .05. Partial correlations control for the other attachment orientation (e.g., the association between attachment anxiety and marriage, controlling for attachment avoidance.

**Beliefs about event-related personality change.** Participants rated how much they thought each of 21 life events would change someone's personality (defined as basic ways of thinking, feelings, and behaving). Events were rated on a 5-point scale: −2(*definitely no*), −1(*probably no*), 0(*maybe*), 1(*probably yes*), and 2(*definitely yes*). The full list of life events (and descriptive statistics) can be found in Table 1.

**Event perceptions.** Participants also completed an 18-item version of the Event Characteristics Questionnaire for two of the hypothetical life events that they were randomly assigned to evaluate [9]. One of the life events was "started college or university." However, because participants were all college students (and thus all started college/university), this event was not rated on event characteristics (because they only rated two randomly-chosen events that they had not experienced before).

The 18-item version provides two items to measure each of the 9 dimensions: challenge ($\alpha$=.82; e.g., "This event was stressful."), change in world views ($\alpha$=.59; e.g., "This event helped me gain new perspectives."), emotional significance ($\alpha$=.72; e.g., "This event moved me a lot."), external control ($\alpha$=.70; e.g., "This event was in the hands of other people."), extraordinariness ($\alpha$=.74; e.g., "Most people like me experience this event sometime in their lives." [reversed]), impact ($\alpha$=.60; e.g., "This event had a strong impact on my life."), (un)predictability ($\alpha$=.87; e.g., "This event occurred suddenly."), social status change ($\alpha$=.76; e.g., "My reputation suffered from this event."), and valence ($\alpha$=.86; e.g., "This event was joyful."). All items were rated using a 5-point scale ranging from 1(*strongly disagree*) to 5(*strongly agree*); see Table 2 for means of each dimension collapsed across life events.

Some of these subscales have relatively low reliabilities, which is likely attributable to our use of the short-form measure of the scale (i.e., two-items per dimension compromises reliability and additional items would provide greater precision). Although these reliabilities are consistent with previous research [9,71], future research can employ the long-form versions of these instruments.

## Analytic approach

For each analysis, we ran models for anxiety and avoidance separately. We first began by correlating attachment orientations with perceptions of how much each life event would change someone's personality.

We then correlated the attachment orientations with overall life event perceptions collapsing across life events (e.g., is attachment anxiety associated with perceiving life events to be more positive or emotionally significant?). Finally, we examined associations between attachment orientations and perceptions by each life event separately to see whether they varied by a particular life event (instead of collapsing across all of them).

**Table 2. Event characteristic descriptives and correlations with attachment orientations.**

| Event characteristic dimension | | | Attachment anxiety | | Attachment avoidance | |
|---|---|---|---|---|---|---|
| | **M** | **SD** | **Raw** | **Partial** | **Raw** | **Partial** |
| Challenge | 4.014 | 0.943 | **0.125** | **0.138** | **−0.082** | **−0.101** |
| Worldview | 3.817 | 0.835 | **0.089** | **0.102** | **−0.086** | **−0.100** |
| Emotional Significance | 4.071 | 0.819 | **0.114** | **0.131** | **−0.109** | **−0.127** |
| Control | 2.736 | 1.042 | 0.005 | 0.001 | 0.024 | 0.023 |
| Extraordinariness | 2.565 | 1.003 | −0.009 | −0.010 | 0.007 | 0.008 |
| Impact | 3.969 | 0.855 | **0.133** | **0.146** | **−0.080** | **−0.100** |
| Predictability | 3.311 | 1.153 | **0.036** | **0.039** | −0.020 | −0.026 |
| Social Status | 2.747 | 1.120 | **0.054** | **0.054** | 0.005 | −0.002 |
| Valence | 2.389 | 1.335 | **−0.043** | **−0.049** | **0.040** | **0.046** |

*Note*. *N*s range from 3845–3847. Bolded correlations are significant at *p*<.05. Partial correlations control for the other attachment orientation (e.g., the association between attachment anxiety and challenge, controlling for attachment avoidance).

Because attachment anxiety and avoidance were positively associated with each other, we also computed partial correlations in which we estimated the association between one attachment orientation (i.e., attachment anxiety) and a life event/perception while controlling for the other attachment orientation. These partial correlations are presented alongside the raw/bivariate correlations. For the most part, the results were highly similar across the two correlations. The associations were typically larger after accounting for the other attachment orientation. There were a few scenarios in which a null bivariate association became significant after controlling for the other attachment orientation. For the Results, we largely discussed the associations that were consistently significant using both approaches. We did this because of the high similarity between the two, the interpretational ambiguity of one attachment orientation being "purged" or unconfounded from the other, and because the few associations that were significant in one approach but not the other were typically very small and unlikely to survive even a modest alpha correction. Nevertheless, they are reported in the tables for transparency.

Data collection was guided by the resources available to us within one semester. Because of the study design, sample sizes varied a bit across analyses. For the perceptions of personality change analyses, these associations are based on the entire sample ($N = 1,928$ as a lower bound, given some missingness; Table 1). Because life events were pooled together (and each participant rated two events), the associations between attachment orientations and event characteristics (pooled) are based on a larger sample ($N = 3,849$, given some missingness; Table 2). Finally, the analyses broken down by each life event were based on a bit smaller samples. Specifically, because each participant rated one life event randomly, these analyses have approximately 190 participants for each correlation (Fig 1 and Fig 2). At 80% power at $\alpha = .05$, we could estimate associations as small as $r = .20$ (for $N = 190$), $r = .06$ (for $N = 1,928$), and $r = .05$ (for $N = 3,849$).

## Results

### Perceptions of life events spurring personality change

As seen in Table 1, the mean was above zero for perceptions of personality change after each life event. These ratings suggest that people believe these events can potentially change a person's personality. The three most significant life

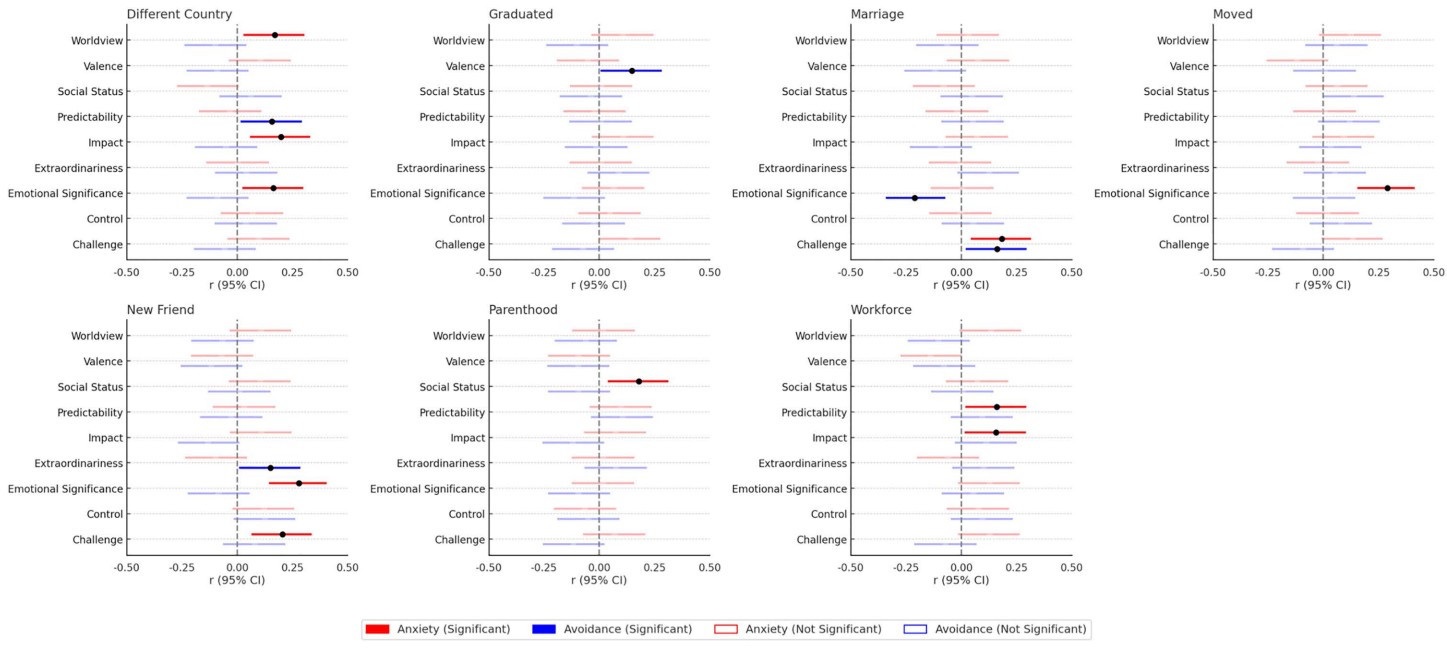

**Fig 1. Associations between attachment orientations and perceptions of positive life events.**

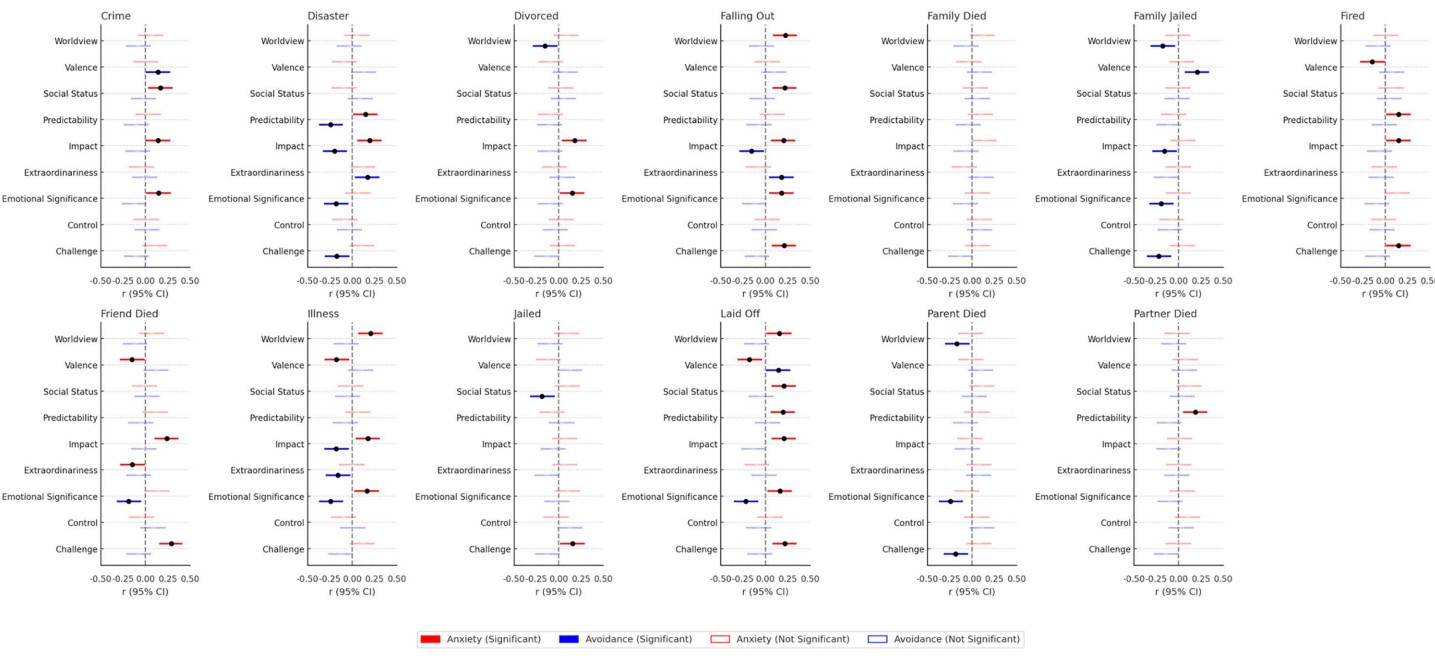

**Fig 2. Associations between attachment orientations and perceptions of negative life events.**

events were having a romantic partner die, having a parent die, and becoming a parent. The three least significant life events were making a new close friend, having a falling out with a friend, and having a close family member jailed or imprisoned (although the means for these were still above the midpoint of zero).

Associations between attachment orientations and personality change perceptions were relatively small. People higher in attachment anxiety tended to think that some life events were particularly likely to cause personality change. Specifically, participants higher in attachment anxiety were more likely to think that getting divorced, being fired/laid off from a job, moving to a new city, having a close friend die, having a falling out with a close friend, starting college/university, being the victim of a serious crime, and experiencing a natural disaster had the potential to change someone's personality.

The pattern of associations with personality change perceptions and attachment avoidance were much larger than those seen for attachment anxiety, although still modest in the absolute sense. For individuals higher in attachment avoidance, the overall pattern is that they did not think any event would likely cause personality changes. This was particularly true for becoming seriously ill/injured, experiencing a natural disaster, being the victim of a serious crime, having a close family member jailed/imprisoned or die, having a romantic partner die, becoming a parent, and spending a significant amount of time in a different country.

### Attachment orientations and event characteristics

Associations between attachment orientations and life event perceptions are reported in Table 2 and were generally small. Across all pooled life events, people higher in attachment anxiety were more likely to think life events would be challenging, change their worldview, be emotionally significant, have a larger impact on them, unpredictable, negatively valenced, and negatively affect their social status.

People higher in avoidance demonstrated many of the opposite effects. Specifically, people higher in attachment avoidance were less likely to think that life events would be challenging, change their worldview, be emotionally significant, impact them, and thought they would be more positively valenced. Attachment avoidance was largely unrelated to the

other event characteristic dimensions. Perceptions of control and extraordinariness were not significantly associated with attachment anxiety or avoidance.

Because participants provided multiple event perception ratings, there is a non-independence issue in the data. We ran follow-up tests predicting each event perception from anxiety or avoidance in the context of a multi-level model (with random intercepts for participants). The results were identical in interpretation as reported in Table 2. The results of these multi-level models can be found in Supplementary S1 Table.

### Attachment orientations and event characteristics for specific life events

Our previous analysis collapsed across all the rated life events. However, doing so likely obscures how attachment orientations are related to perceptions of *particular* life events. To more succinctly summarize the results, we divided the 20 life events into ostensibly positive and negative life events [see 71, for a discussion of this classification]. As a reminder, because participants were college students, we did not examine starting college/university as a hypothetical life event. We plotted the results for attachment anxiety (red markers) and attachment avoidance (blue markers) within the same forest plots. Non-significant correlations and their confidence intervals are faded. For this information in table form, for both bivariate and partial correlations, see Supplementary S2 Table. Comparing the bivariate and partial correlations revealed that only about 5% of the associations changed in significance level (3 went from significant to non-significant; 17 went from non-significant to significant), although these were typically just below $p = .05$ and would be unlikely to survive an alpha correction; see the supplement for full details.

Associations between attachment orientations and perceptions of positive life events were generally weak and inconsistent (see Fig 1). Of the 126 correlations tested, only 15 (11.9%) reached statistical significance, which would be further reduced with even a modest alpha correction. Individuals higher in attachment anxiety tended to view certain positive events as more emotionally charged or impactful. For example, they perceived entering the workforce and moving to a different country as especially consequential, and they rated marriage and making a new friend as particularly challenging. They also described making a new friend as an extraordinary life event and considered moving a long distance or spending time abroad to be more emotionally significant. Anxiously attached individuals were also more likely to believe that becoming a parent would harm their social standing, that entering the workforce would be unpredictable, and that spending time in another country could change their worldview.

In contrast, individuals higher in attachment avoidance showed even fewer associations with positive event perceptions. Participants higher in attachment avoidance perceived marriage as less emotionally significant but more challenging, and they viewed spending time in a different country as more unpredictable. They were also more likely to see making a new friend as extraordinary and graduation as a particularly positive experience. Overall, however, there was little evidence that attachment orientations meaningfully shaped how people anticipated responding to positive life events.

Patterns were somewhat stronger for negative life events, although still modest (see Fig 2). Of the 234 correlations tested, 57 (24.4%) were statistically significant, which would be further reduced with even a modest alpha correction. Participants higher in attachment anxiety were more likely to perceive negative events as especially disruptive and emotionally significant. They believed that experiences such as a falling out with a friend, a serious illness, or being laid off could meaningfully alter their worldview. They also viewed events like being fired, losing a friend, becoming seriously ill, or being laid off as highly negative and likely to harm their social standing. These individuals tended to perceive natural disasters and job loss as unpredictable, and they judged a broad range of events—including victimization, divorce, and illness—as particularly emotionally significant, impactful, or challenging. Interestingly, they also perceived a friend's death as a relatively ordinary event, suggesting a more complex or possibly numbed perception of interpersonal loss.

Those high in attachment avoidance reported fewer associations overall. They were less likely to believe that events like divorce, a parent's death, or a family member being jailed would change their worldview. Individuals higher in attachment avoidance rated some negative events—such as victimization, being laid off, and having a family member jailed—as

less negative than less avoidant individuals, and they were also less likely to think that being jailed would damage their social status. They perceived certain events (e.g., natural disasters and serious illness) as more predictable, and generally viewed a range of negative experiences as less emotionally significant and less challenging compared to those lower in attachment avoidance.

In summary, while attachment anxiety and avoidance were associated with select perceptions of both positive and negative life events, these patterns were inconsistent and generally small in magnitude. The findings suggest that attachment orientations may shape people's expectations about life events in limited and context-dependent ways, particularly when considering hypothetical scenarios.

## Discussion

In a large sample, we examined attachment-related differences in perceptions of 20 different life events in a large sample. Although the effects were generally small, a few consistent patterns emerged. First, people high in attachment avoidance largely thought that their personalities would be less affected by both positive and negative life events; people high in attachment anxiety thought their personalities would change more dramatically, although attachment anxiety was not as robustly related to personality change perceptions. We found that attachment anxiety was associated with perceiving life events as more negative, more challenging, more emotionally significant, more impactful, less predictable, damaging to their social standing, and would be more likely to change their worldviews. People higher in attachment avoidance felt the opposite—that life events were not as challenging, less emotionally significant, less impactful, more positive, and unlikely to change their worldviews. Examining perceptions of particular life events yielded findings that were mostly consistent with these overall associations, but with very few consistent effects. These findings extend previous work on life event perceptions to the study of adult attachment orientations and provide a preliminary roadmap for understanding how and why life events do or do not change attachment orientations.

To contextualize these results, the associations reported here were very small in magnitude, even though they were statistically significant. This pattern likely stems from our large sample of participants who provided multiple life event perceptions, as smaller samples often overestimate effects sizes [82]. Are these small-but-significant associations of *practical* significance though? When asking about event perceptions in a hypothetical and decontextualized way, there are likely many contributing factors that affect the magnitude of these associations. Although our focus was on attachment orientations, for which we found some patterns consistent with previous work, the particular impact of these factors in more specific contexts is something our study did not examine. At such a broad level, one could characterize the results of the current study has small and, in some cases, negligible. However, it is possible that the influence of attachment orientations might be enhanced or reduced depending on other aspects of the context (e.g., real/actual life events, other characteristics of the life event or participants). Thus, although the associations found here are relatively small, their impact across people and settings might be more heterogeneous than what is reflected here. In other words, attachment orientations might be more influential in describing life event perceptions for some people than others (or as life events or social contexts and experiences accumulate). Below, we contextualize our results in the existing literature. However, we encourage caution in evaluating these results given the relatively small effect sizes.

### Attachment orientations and how cognitive and motivational tendencies reify them

The present findings extend adult attachment theory by demonstrating that people's perceptions of life events map meaningfully onto the cognitive and motivational tendencies that underlie attachment anxiety and avoidance [20,32,36,83]. Rather than reflecting random variation or general negativity, these patterns appear to be consistent with stable working models that guide how individuals attend to, interpret, and emotionally respond to potentially meaningful events. In line with theoretical frameworks emphasizing the biased cognitive processing in insecure attachment, our results suggest that people higher in attachment anxiety and avoidance do not simply perceive hypothetical life events differently, but they

perceive and evaluate those circumstances in ways that reinforce their existing relational beliefs. Thus, rather than serving as corrective experiences, life events may function psychologically as confirmatory evidence that maintains—rather than disrupts—attachment orientations [26,28,30,31,84–86].

Individuals higher in attachment anxiety appeared particularly likely to interpret life events as impactful, emotionally significant, and potentially destabilizing, consistent with hyperactivating strategies that prioritize vigilance, emotional amplification, and threat sensitivity. Their greater belief that events could meaningfully alter personality may reflect an underlying sense of vulnerability and uncertainty about relational stability [17,47]. Given their tendency to scrutinize relational cues and interpret ambiguous information as threatening or rejection-relevant [30], anxious individuals may perceive life events—especially those involving transition or uncertainty—as opportunities for potential loss, change, or validation of their feared expectations [76]. This aligns with work showing that individuals high in anxiety engage in rumination, catastrophizing, and selective attention toward negative relational cues, which may contribute to the perception that life experiences have powerful and lasting negative consequences [87]. Taken together, the current findings suggest that anxious individuals' heightened emotional and cognitive engagement with life events may increase the likelihood that such events shape their relational beliefs over time.

Attachment avoidance was more robustly associated with personality change perceptions. Conversely, though, people higher in attachment avoidance perceived nearly all life events as low in personal relevance and unlikely to influence their personality. This pattern is consistent with deactivating strategies aimed at minimizing reliance on others, suppressing emotional engagement, and maintaining psychological distance [88–90]. Avoidant individuals' tendency to downplay the relational and emotional significance of events may serve as a defensive mechanism that protects their preferred autonomy and reduces exposure to vulnerability. Prior research demonstrates that avoidant adults often exhibit attentional disengagement, reduced emotional recall, and reinterpretation of relational experiences as inconsequential [44,54,55]. All of these behavioral tendencies may contribute to the belief that external circumstances—including potentially disruptive or meaningful ones—pose little threat to their self-concept. In this way, avoidance may buffer individuals from perceived relational instability but, paradoxically, may also prevent them from using life experiences as opportunities for growth or relational recalibration.

## Attachment and perceptions of life events

The taxonomy for characterizing perceptions of life events used here was recently developed to potentially explain why some life events lead to enduring changes in psychological characteristics [9]. These perceptions range from emotional evaluations of life events (e.g., valence, impact, challenge, emotional significance), to attempts to understand their causes (e.g., control, extraordinariness, predictability), to perceptions of how they might lead to psychological changes (e.g., worldview changes, social status changes). Recent work provides some evidence that assessing perceptions of life events (instead of just whether a life event occurred or not) is a fruitful endeavor, as they explain variation in how people's personality and well-being change following life events [66–69]. Individual differences in Big Five personality traits are associated with these perceptions [71]. And, although a lot of relatively disconnected evidence has linked attachment orientations to particular types of perceptions [e.g., locus of control; 91], we took a more systematic approach to examine attachment-related differences in a variety of life event perceptions.

Indeed, attachment orientations were associated with perceptions of life events. Attachment anxiety was associated with perceiving life events as more challenging, more likely to affect their worldview, more impactful, more emotionally significant, more negative, less predictable, and more damaging to their social status. Collectively, all of these perceptions are consistent with the observation that anxiously attached adults respond more intensely and negatively than less anxious adults. Knowing that individuals high in attachment anxiety have more intense perceptions is consistent with their endorsement that life events would likely change their personalities. It is also consistent with work showing that life events do tend to affect changes in attachment anxiety more than attachment avoidance. The negative filtering of even ostensibly

positive life events in our life-event-specific analysis is also noteworthy in that, even when positive events happen, people higher in attachment anxiety tend to view them a little more negatively than people lower in attachment anxiety [29]. A close examination of the longitudinal study by Fraley and colleagues [7] also shows that some ostensibly positive relationship events (e.g., getting engaged, having a partner do something nice) are associated with longer-term *increases* in attachment anxiety. The authors note that evaluating life events as more positive tended to be associated with declines in attachment anxiety and that some of these events might reflect a compensatory action done on behalf of partners (e.g., the relationship is going poorly, so one partner tries to do something nice for the other; but this poor relationship might explain why attachment anxiety is higher). Our findings here help further contextualize the effect of life events on attachment. Specifically, people might not only evaluate an event as more positive or negative. Rather, people higher in attachment anxiety offer a whole host of reasons why life events are not exclusively positive—they also view them as challenging, disrupting their worldview, impactful, and damaging to their social standing.

An extension of this line of reasoning can be provided for attachment avoidance. Essentially, people higher in attachment avoidance tended to minimize their evaluation of life events, suggesting they were less challenging, unlikely to change their worldviews, less emotionally significant, less impactful, and more positive. Such minimization is consistent with the tendencies of people higher in attachment avoidance to deploy deactivating strategies in the face of threatening information [32,48,53]. And, indeed, attachment avoidance seems to be less susceptible to life-event-related changes [7]. For the few times in which life events exerted longer-term influences, they tended to do so in ways that reify avoidance: marriage was associated with increases in attachment avoidance, and dissolutions were associated with declines in attachment avoidance. Future research can examine whether individual differences in changes and avoidance trajectories following a life event (which were present in the Fraley et al. study) might be explained by these other perceptions.

## Limitations and future directions

The current study had many strengths. We examined attachment-related differences in perceptions of life events using a broad range of positive and negative life events. We are also among the first to examine a comprehensive set of event perceptions—useful information for the field moving forward to characterize how life events might engender psychological change. Nevertheless, there are limitations that should be acknowledged.

First, we relied on cross-sectional data and hypothetical life events. Appropriately examining how life events change a person's attachment orientation requires longitudinal data, including assessments of attachment orientations before and after an event [e.g., 7]. Unfortunately, data sets that have multiple assessments of attachment orientations over time often do not have comprehensive assessments of life events or their perceived characteristics (nor do studies of life events [and their characteristics] measure attachment very often). In other words, researchers might have information on whether a life event happened and occasionally multiple assessments of attachment. Still, they lack description and specificity of how life events were perceived or affected people. Our study suggests that there were attachment-related differences in how people perceived *hypothetical* life events, suggesting that these perceptions might have differential effects on how attachment changes when people actually experience events. However, there is no guarantee that these hypothetical perceptions extend to real events. In this way, our study was limited in its ability to make causal claims for characterizing how attachment orientations would be related to evaluations of *actual* life events and experiences, or how how people might adapt to actual life events that happen to them. Future research should more deliberately integrate information about how participants subjectively perceive the impact of actual—not hypothetical—life events along a broad array of indicators [9]. These studies would enable us to test whether these perceptions (and not just the presence or absence of a life event) lead to changes in attachment orientation.

Second, our participants were from one particular culture (i.e., the United States). Most psychological research is conducted in predominantly WEIRD (Western, educated, industrialized, rich, democratic) countries—a considerable limitation to the current work. There are certainly studies that have examined variation in attachment across cultures [92–96]. Given

that demographic variation in attachment orientations differs across regions and cultures, it seems likely that how people perceive and evaluate life events also differs [94,97], for which there is evidence [98,99]. However, the study of life events, both with respect to attachment and psychological characteristics more broadly, has been predominantly done in WEIRD cultures [8,100]. It is an important future direction to elevate the study of life events (and their perceptions), attachment, and their intersection into the cross-cultural space. Just as attachment differs across cultures [94], event perceptions and their impact on attachment changes across the lifespan might also vary.

Third, most of the associations reported here are very small in magnitude. Because the sample size was relatively large (particularly for some of the pooled analyses), many associations were statistically significant despite there being a question of whether they are meaningfully significant. In addition to some of those associations being near-zero, their statistical significance would be questioned with even a modest correction for multiple testing. Despite some precedent from past work that would suggest modest effect sizes, attachment orientations were not strongly related to event perceptions despite some of the results being consistent with past research. Future research can more appropriately contextualize the role of perceptions in changes in attachment following life events. It is possible that small associations between attachment orientations and perceptions (measured cross-sectionally using hypothetical events) could be different when examining these questions in longitudinal data using real life events.

Finally, our sample was comprised primarily of women and younger adults. It is possible that the gender and age composition of the sample might have altered the magnitude of some of our associations. Women tend to report higher attachment anxiety, and men tend to report higher avoidance [101], although the magnitude of these differences can be smaller among middle-age and older adults [6]. As a result, our sample was likely biased toward characterizing young women's experiences in particular. And men and women often have unique responses to relational circumstances [102,103]. Thus, the large number of women might have yielded a sample higher in attachment anxiety (and lower in attachment avoidance), on average. This bias was likely compounded by the primary inclusion of younger adults. There are normative shifts in attachment anxiety across the lifespan, such that attachment anxiety is higher among younger adults, although age differences in attachment avoidance are a bit more mixed [3,6,21,92,104–106]. Likewise, there are also age-related differences in event perceptions, both in terms of which events are more common and how they are evaluated according to Luhmann et al.'s taxonomy [107]. Specifically, health-related events are more common among older adults and education-related events are more common among younger adults. Further, young adults perceived life events more positively and as more likely to change their worldviews. Perceptions were often more intense if experienced at a non-normative age. Altogether, the evidence that attachment and event perceptions vary across age and gender suggests that future research should strive to collect more age- and gender-balanced samples moving forward.

## Implications

The current work has important clinical and theoretical implications. For example, many existing clinical psychology frameworks suggest that enduring vulnerabilities and diatheses, when paired with particular external circumstances, result in the development of interpersonal difficulties [39,108–110]. Perceiving an event as more negatively valanced, challenging, extraordinary, and negatively affecting one's social status are associated with declines in life satisfaction after that life event [70]. Attachment orientations could be considered a pre-existing characteristic that could potentially shape how people perceive life events or how these perceptions translate to adaptation following the life event. For example, although the associations were small, people high in attachment anxiety perceived life events as more challenging, negatively valanced, and more likely to negatively impact their social status. In a therapeutic context, this information could be valuable in assessing how people might navigate important life shocks and to develop intervention plans catered to the concerns of people high in attachment anxiety. People high in attachment avoidance tended to minimize the emotional significance of life events. Although this might initially be useful in reducing stress and anxiety, such distancing might

inadvertently prevent adaptive processing of life events' impact on people or stymie therapeutic efforts to change the everyday behavior of people high in attachment avoidance.

Most broadly, from a theoretical perspective, situating attachment orientations in an emergent event perceptions classification system provides a unifying framework for understanding attachment-related differences in responses to life events. Overall, life events exert little effect on attachment orientation [7]. The lack of effects seen in previous research might be due in part to the missing context of how people were perceiving and evaluating those life events. For example, people in attachment anxiety have a propensity to catastrophize which was illustrated by their perception of hypothetical events as more challenging, impactful, and able to negatively affect their social status. Evaluating life events in these ways might reify the internal working models and attributional tendencies of people high in attachment anxiety—one of the suggested reasons why psychological characteristics are so stable over time [26]. The same can be said for attachment avoidance. By downplaying the impact of life events, people high in attachment avoidance might be reifying their defensive perceptions of others and the world. But this consistency likely comes at a cost: by not properly evaluating life events, particularly relational ones, people high in attachment avoidance might be less likely to alter their behavior to salvage a relationship, reflect on a disruptive experience, or seek out support from others. In both of these cases, attachment orientations filter one's experience and reinforce the behavior consistent with their attachment orientation. Having a broader understanding of life event perceptions reveals the exact ways that these filters might exert their influence in the lives of people high in attachment anxiety and avoidance.

## Conclusion

The current study examined attachment-related differences in life event perceptions across 20 different life events. Although the associations were small, attachment anxiety was positively associated with thinking that life events would change their personality, but attachment avoidance was negatively associated with thinking life events would change their personality. People higher in attachment anxiety and avoidance often held opposing perceptions of life events: People high in attachment anxiety thought life events were more challenging, likely to change their worldview, were emotionally significant, impactful, and likely to negatively impact their social status. People high in attachment avoidance generally felt the opposite. Analysis of particular life events revealed few consistent findings beyond these general patterns. The current study adds important context for evaluating how life events are perceived differently depending on attachment orientations. Our hope is that the results presented here may motivate future studies to examine whether differences in life event perceptions might partially explain attachment-related changes across the lifespan.

## Supporting information

**S1 Table. Associations between event characteristic descriptives and attachment orientations in the context of multi-level modeling.**
(DOCX)

**S2 Table. Associations between attachment orientations and life event perceptions (By life event).**
(DOCX)

## Author contributions

**Conceptualization:** Logan C. Gibson, William J. Chopik.

**Data curation:** William J. Chopik.

**Formal analysis:** Logan C. Gibson, William J. Chopik.

**Investigation:** William J. Chopik.

**Methodology:** Logan C. Gibson, William J. Chopik.

**Project administration:** William J. Chopik.

**Resources:** William J. Chopik.

**Supervision:** William J. Chopik.

**Validation:** William J. Chopik.

**Visualization:** Logan C. Gibson, William J. Chopik.

**Writing – original draft:** Logan C. Gibson, William J. Chopik.

**Writing – review & editing:** Logan C. Gibson, William J. Chopik.

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
