## [Decision Letter · Decision Letter 0]

7 Oct 2025

PONE-D-25-36404

Attachment-related Perceptions of Life Events

PLOS ONE

Dear Dr. Chopik,

Thank you for submitting your manuscript to PLOS ONE. After careful consideration, we feel that it has merit but does not fully meet PLOS ONE’s publication criteria as it currently stands. Therefore, we invite you to submit a revised version of the manuscript that addresses the points raised during the review process.

We look forward to receiving your revised manuscript.

Kind regards,

Tobias Otterbring, PhD Psychology

Academic Editor

PLOS ONE

Journal Requirements:

Additional Editor Comments :

Dear Prof. Chopik,

Your manuscript was sent to three experts in this topic domain. I have now received their feedback. All the reviewers like the idea of your manuscript and jointly recommend you undertake a major revision (their detailed comments appear below or in separate documents). However, they also note a set of substantial concerns related to the conceptualization and analytic approach as well as various issues related to generalizability, realism, interpretation of effect sizes beyond statistically significant results, lack of clarity in theorizing, and inconsistent use of certain central terms. Based on my own reading of the manuscript, I largely concur with the constructive comments from the reviewers. Therefore, my recommendation as the Associate Editor is to invite you for a major revision, with a relatively clear path toward publication as long as you carefully address all the issues brought up by the reviewers. Please submit your revision at your earliest convenience and no later than 3 months after having received this decision. If you need additional time, please let me know (at tobias.otterbring@uia.no). Good luck with the revision!

Kind regards,

Tobias Otterbring, PhD Psychology

Professor of Marketing

Associate Editor, PLOS One

Reviewers' comments:

Reviewer's Responses to Questions

**Comments to the Author**

1. Is the manuscript technically sound, and do the data support the conclusions?

Reviewer #1: Yes

Reviewer #2: No

Reviewer #3: Yes

2. Has the statistical analysis been performed appropriately and rigorously? 

Reviewer #1: Yes

Reviewer #2: No

Reviewer #3: Yes

3. Have the authors made all data underlying the findings in their manuscript fully available?

Reviewer #1: Yes

Reviewer #2: Yes

Reviewer #3: Yes

4. Is the manuscript presented in an intelligible fashion and written in standard English?

Reviewer #1: Yes

Reviewer #2: Yes

Reviewer #3: Yes

5. Review Comments to the Author

Reviewer #1: Dear Authors,

Thank you for the opportunity to review this interesting and well-executed manuscript. You have undertaken a commendable study on a theoretically rich and important topic, exploring how attachment orientations shape the perceptual lens through which life events are viewed. The application of Luhmann et al.’s taxonomy is a particular strength. I have several suggestions that I believe could help strengthen the theoretical contribution and clarity of the paper.

Major Comments:

1. Deepening the Theoretical Framework: The “Why” Question

The introduction and discussion would benefit greatly from moving beyond describing the differences to explaining them. The findings are a perfect springboard to discuss the underlying motivational and cognitive mechanisms of attachment strategies.

Attachment Anxiety: The results suggest anxious individuals perceive events as highly impactful, negative, and identity-altering. This should be explicitly framed as a consequence of their hyperactivating strategies. Their chronic need for proximity and fear of abandonment creates a cognitive-affective framework where events are intensely scrutinized for their relational meaning and potential threat. This aligns with their need to create a coherent narrative (often negative) that justifies their anxiety and motivates their search for reassurance.

Attachment Avoidance: The dismissive perceptions of avoidant individuals are a textbook example of deactivating strategies. Their defensive minimization of event significance, emotional impact, and potential for change is a core strategy to maintain independence and suppress attachment-related distress. This isn't just a "difference"; it's a strategic downregulation of the perceived importance of events to avoid the vulnerability that comes with acknowledging their impact.

I strongly recommend restructuring the discussion around these theoretical frameworks (hyperactivation/deactivation) to answer the "why" question more compellingly.

2. Nuancing the Link to Personality (Big Five)

The authors briefly mention the work on Big Five traits but do not fully integrate this into their narrative. This is a missed opportunity. Given the well-established correlations (Attachment Anxiety ~ Neuroticism; Attachment Anxiety and Attachment Avoidance ~ low Agreeableness), the current findings could be partly explained by these broader traits.

For instance, is the heightened negative perception in anxious individuals merely a reflection of their neuroticism? The discussion should explicitly address this potential confound and argue for the unique contribution of attachment working models over and above the Big Five. A control analysis (e.g., partial correlations controlling for Neuroticism) would be a powerful addition, though even a theoretical discussion of this point is necessary.

3. Interpretation of Weak Effects and Gender Imbalance

The authors correctly note the correlations are weak. This warrants a more critical discussion. Do these small effects suggest that while statistically discernible in a large sample, attachment orientation is a relatively minor player in event perception compared to other factors (e.g., the specific event itself, broader personality)? The practical significance of these findings needs to be addressed.

The sample is 74% women. Given established gender differences in attachment (higher anxiety in women, higher avoidance in men), this imbalance likely skewed the data. The authors must discuss how this may have affected the results. For example, did the preponderance of women inflate the mean levels of anxiety and the effects related to it? Would the avoidance-related effects be stronger in a more gender-balanced sample? This is a significant limitation that should be discussed in depth.

4. Terminology and Clarity

Please use consistent terminology throughout the manuscript. Phrases like "insecurely attached people," "people high in attachment avoidance," and "avoidantly attached people" are used interchangeably. For clarity and precision, I recommend standardizing to the dimensional language: "individuals high in attachment anxiety" and "individuals high in attachment avoidance."

5. Implications Section

The discussion would be strengthened by a dedicated paragraph on the implications of these findings. For example:

Clinical Implications: Understanding these perceptual biases can inform therapy. Helping an anxious client recognize their tendency to catastrophize events, or an avoidant client to acknowledge their minimizing strategies, could be a fruitful therapeutic target.

Theoretical Implications: The findings support a model where attachment styles are not just outcomes but active filters that shape how experiences are construed, which in turn may reinforce the existing attachment style, creating a self-perpetuating cycle.

Conclusion:

This is a valuable study with solid methodology that addresses an important question. With revisions that deepen the theoretical explanation, acknowledge limitations more critically, and clarify the narrative, it has the potential to be a strong contribution to the literature.

Reviewer #2: The paper addresses an important and timely question: how individuals with different attachment orientations perceive and evaluate significant life events. The topic is highly relevant for understanding both personality development and subjective interpretations of change-inducing experiences. The study is well-powered and offers a broad empirical dataset. Nevertheless, I have several major concerns regarding both the framing of the paper and the analytic strategy.

1. According to the title, the abstract, and the main body of empirical work, the central focus of this article is how attachment orientations influence perceptions of life events — whether events are viewed as challenging, impactful, emotionally significant, unpredictable, negative, or likely to induce personality change. From this perspective, it is somewhat surprising that the theoretical section devotes substantial space to discussing whether attachment orientations themselves change over time. This issue, while relevant to the general literature on attachment, does not appear to be directly aligned with the empirical aims of the present study. Instead, the introduction would have benefited from a more focused review of how attachment orientations may shape the perception and appraisal of life events.

2. In reporting the results, the authors analyze correlations separately for attachment anxiety and attachment avoidance. However, prior research consistently shows that these two dimensions are positively intercorrelated. As a consequence, some of the reported associations may be confounded by this interrelation. A more appropriate analytic strategy would be to present partial correlations controlling for the other attachment dimension, or, preferably, to use regression models in which both attachment anxiety and avoidance are entered simultaneously as predictors of each perception rating. This would provide a clearer and more valid picture of the unique contributions of each attachment dimension.

3. The manuscript highlights that individuals higher in attachment anxiety tend to perceive certain events (e.g., divorce, job loss, relocation) as more likely to induce personality change. However, the actual correlation coefficients for these examples are on the order of r = .05–.06. Such effect sizes explain only a negligible fraction of variance, and the statistical significance is a function of the very large sample size rather than substantive explanatory power. By contrast, the associations with attachment avoidance, while still modest (rarely exceeding r = .15), are consistently stronger than those observed for attachment anxiety. Yet the discussion does not adequately acknowledge this pattern, giving the impression that anxiety plays the dominant role when the data suggest the opposite.

4. In the section Attachment Orientation and Event Characteristics, the authors pool all events and compute correlations between attachment dimensions and event ratings across approximately 3,800–3,900 observations. However, these observations are not independent, as each participant provided ratings for multiple events. Attachment anxiety and avoidance are measured at the person level, whereas challenge, worldview change, emotional significance, and other characteristics are measured at the event level, nested within participants. Simple correlations therefore ignore the non-independence of the data and likely inflate the effective sample size. A more appropriate analytic approach would be multilevel modeling (e.g., mixed-effects regression), with events nested within persons, which would allow for proper estimation of effects while accounting for shared variance at the participant level.

5. In the section Attachment Orientations and Event Characteristics for Specific Life Events, the authors again present separate correlations for attachment anxiety and attachment avoidance, further split into positive and negative life events. This approach suffers from the same limitation noted above: it does not control for the correlation between anxiety and avoidance, making it unclear which associations are unique to each attachment dimension. As before, a regression approach including both dimensions simultaneously would provide more interpretable and theoretically meaningful results.

Reviewer #3: Thank you for allowing me to review your study. I found it quite engaging and valuable, especially since it tackles an important gap in attachment research by exploring the link between attachment orientations and perceptions of life events.

I identified some theoretical, methodological, and interpretive issues that somewhat limit its overall contribution.

How did you approach investigating attachment in relation to how people perceive subjective events, beyond studies that only examine the occurrence of events?

Authors should also emphasize the importance of integrating attachment theory with emerging event classification systems.

From a methodological standpoint, relying on U.S. undergraduates (average age = 19.6, 74% women) restricts the generalizability of the findings; while the WEIRD bias is acknowledged, it remains a major concern.

Another issue is the cross–sectional design based on hypothetical scenarios. I believe that perceptions of imaginary events may not accurately reflect real experiences, and making causal claims about how life events influence attachment is questionable.

In terms of data analysis, some ECQ subscales (e.g., worldview, α = 0.59, and impact, α = 0.60) exhibit low reliability.

Additionally, most identified relationships are very weak between the studied constructs. Although statistically significant, these limitations raise doubts about their practical significance.

Some conclusions, such as the resilience of avoidant individuals and the greater reactivity of anxious individuals, might overstretch the small correlational data.

6. PLOS authors have the option to publish the peer review history of their article (what does this mean? ). If published, this will include your full peer review and any attached files.

**Do you want your identity to be public for this peer review?** For information about this choice, including consent withdrawal, please see our Privacy Policy .

Reviewer #1: **Yes: ** Emrullah Ecer

Reviewer #2: No

Reviewer #3: No

---

## [Author Response · Author response to Decision Letter 1]

22 Nov 2025

see uploaded document for appropriate formatting

Response to Reviewers

We would like to thank the editor and reviewers for their thoughtful comments on the manuscript. We very much appreciate the constructive feedback and believe that the manuscript has improved significantly as a result of their suggestions. Below, we report how each of the reviewer issues was addressed and the corresponding changes to the manuscript (we also submitted a tracked changes version of the paper). The reviewer comments are non-bolded, and our responses are bolded.

Reviewer #1

Dear Authors,

Thank you for the opportunity to review this interesting and well-executed manuscript. You have undertaken a commendable study on a theoretically rich and important topic, exploring how attachment orientations shape the perceptual lens through which life events are viewed. The application of Luhmann et al.’s taxonomy is a particular strength. I have several suggestions that I believe could help strengthen the theoretical contribution and clarity of the paper.

Major Comments:

1. Deepening the Theoretical Framework: The “Why” Question

The introduction and discussion would benefit greatly from moving beyond describing the differences to explaining them. The findings are a perfect springboard to discuss the underlying motivational and cognitive mechanisms of attachment strategies.

Attachment Anxiety: The results suggest anxious individuals perceive events as highly impactful, negative, and identity-altering. This should be explicitly framed as a consequence of their hyperactivating strategies. Their chronic need for proximity and fear of abandonment creates a cognitive-affective framework where events are intensely scrutinized for their relational meaning and potential threat. This aligns with their need to create a coherent narrative (often negative) that justifies their anxiety and motivates their search for reassurance.

Attachment Avoidance: The dismissive perceptions of avoidant individuals are a textbook example of deactivating strategies. Their defensive minimization of event significance, emotional impact, and potential for change is a core strategy to maintain independence and suppress attachment-related distress. This isn't just a "difference"; it's a strategic downregulation of the perceived importance of events to avoid the vulnerability that comes with acknowledging their impact.

I strongly recommend restructuring the discussion around these theoretical frameworks (hyperactivation/deactivation) to answer the "why" question more compellingly.

Thanks for recommending this revised structure. We agree that it provides a closer introduction to the empirical tests we conducted. We have now significantly changed the introduction—removing the large amount of text on attachment-related changes (which we now realize was a bit tangential). Rather, we took the opportunity, as the reviewer suggested, to dedicate more space to the cognitive and motivational tendencies of people high in attachment anxiety and avoidance. These revisions can be seen on lns 42-125 and are reproduced below:

An individual’s attachment orientation is how they generally approach close relationships and is thought to be formed by early interactions with caregivers, though experiences with relationships later in life likely also affect attachment orientations [1, 2, 10-15]. Variation in attachment orientation is characterized by how people vary on two dimensions—attachment anxiety and attachment avoidance [16]. Attachment anxiety is characterized by fear of rejection and a preoccupation with seeking reassurance from close others [17]. Attachment avoidance is characterized by a discomfort with closeness and dependence [18]. Since its extension from childhood to adult romantic and platonic relationships, attachment theory has served as an organizing framework for characterizing thoughts, feelings, and behavior in the context of close relationships [19, 20]. People low in both attachment anxiety and attachment avoidance are considered to be relatively securely attached.

Although there are normative shifts in attachment orientations across the lifespan, attachment orientations are generally considered stable individual difference characteristics [12, 21-23]. One of the reasons why attachment orientations are stable is because of the cognitive and motivational mechanisms that reify one’s existing “working model” of attachment to promote stability [4, 11, 24, 25]. In other words, like other psychological characteristics, attachment orientations guide the situations people find themselves in, how they interpret those situations, and can also fundamentally change aspects of those situations [26, 27]. People higher in attachment anxiety and avoidance tend to interpret situations in ways that are consistent with what they think about themselves and other people [24, 28-31].

For example, people higher in attachment anxiety tend to engage in hyperactivation of the attachment system as a way of interpreting the world and regulating their emotions and behavior. Attachment-related threats typically trigger efforts to increase proximity to attachment figures and to secure support from them [32, 33]. Collectively, these strategies are meant to prevent abandonment [34]. People high in attachment anxiety tend to hold negative perceptions of themselves, think their self-worth is conditional, and have heightened rejection sensitivity to what they feel are potentially unavailable attachment figures [35-37]. Hyperactivation of the attachment system is accompanied by a hyperfixation on the availability of close others, intense scrutiny over support provision, selective attention toward rejection-related cues, and constant threat-monitoring [38-41]. Among people higher in attachment anxiety, this hyperactivation is often made manifest in intense feelings of jealousy (a vigilance to the attention and commitment of others), maladaptive attributions of ambiguous behavior (assuming others are inconsistently reliable as a way of monitoring availability), and otherwise perceiving the world in a more negative light, as this is consistent with their internal way of thinking about the world [30, 42, 43]. Although these attributional and motivational tendencies make sense in light of their attachment histories and working models, this hyperactivation of the attachment system often results in greater error in threat detection, an ironic undermining of relational commitment from partners, rumination and catastrophizing, and an inaccurate warping of relational circumstances—all correlates of less satisfying interpersonal relationships [17, 44-47]. As a result, when confronted with either an ambiguous life event or situation, people high in attachment anxiety would likely perceive it to be more negative for them, their relationships, and their future.

In contrast, people higher in attachment avoidance engage in chronic attempts to deactivate the attachment system. When attachment needs are triggered, people high in attachment avoidance typically engage in strategies designed to suppress dependency, dampen emotional activation, and maintain distance from attachment figures [18, 34, 48, 49]. Collectively, these strategies are meant to prevent vulnerability and protect autonomy rather than pursue closeness [32, 50]. People high in attachment avoidance tend to hold a negative perception of close others—often believing others are unreliable, intrusive, or emotionally overwhelming—while maintaining a compensatory positive self-view centered on independence and self-sufficiency [35, 51, 52]. Deactivation of the attachment system is accompanied by selective attention away from emotional cues, minimizing the personal relevance of relational events, and the suppression or dismissal of attachment-related thoughts and feelings [53-55]. Among people higher in attachment avoidance, this deactivation often manifests in emotional disengagement, reluctance to self-disclose, downplaying relational difficulties, and attributing ambiguous situations in ways that preserve psychological distance and self-reliance [40, 56, 57]. Although these tendencies serve an immediate regulatory function by reducing perceived relational threat, they may also result in reduced emotional awareness, impaired intimacy, and difficulty acknowledging support when it is offered—patterns that predict lower relationship satisfaction and reduced partner responsiveness over time [58-61]. As a result, when confronted with an ambiguous event or situation, people high in attachment avoidance would likely perceive it as less personally or relationally relevant and respond with emotional disengagement rather than threat sensitivity.

In these ways, people high in attachment anxiety or avoidance have several cognitive and motivational tendencies that shape how they perceive themselves, others, and the world. And these tendencies—and the perceptions they shape—likely explain why people’s attachment orientations are stable, even when disruptive and significant life shocks or events happen to them [62-64], even if some relational contexts might alter these enduring processes [5, 60, 65]. But the study of life events and their impact on attachment changes has been limited in several ways [8]. As an illustration of just how powerful these mindsets are, Fraley and colleagues [7] followed over 4,000 people over a mean time frame of 2 years to examine how their attachment orientations changed before-to-after 25 different life events that encapsulated both relational and non-relational contexts. Although many life events had an initial association with changes in attachment orientations, very few had enduring effects. Importantly, this work rarely examines people’s qualitative assessments of life events—a potential avenue through which attachment orientations might exert their influence. In the current study, we examined how people high in attachment anxiety and avoidance perceive life events as an illustrative case to assess their cognitive and motivational tendencies—tendencies that explain sources of anxiety and avoidance and why they might be durable to changes in situations and life events. But comprehensive models for conceptualizing the range of perceptions people have in response to events are rare. However, researchers have recently developed a taxonomy for characterizing people’s perceptions of life events.

We also revised one of the early parts of the Discussion to discuss how the personality-change-perceptions results are consistent with the aforementioned motivational and cognitive mechanisms outlined in the intro. This addition can be found in lines 446-486 and is reproduced here:

The present findings extend adult attachment theory by demonstrating that people’s perceptions of life events map meaningfully onto the cognitive and motivational tendencies that underlie attachment anxiety and avoidance [20, 32, 36, 82]. Rather than reflecting random variation or general negativity, these patterns appear to be consistent with stable working models that guide how individuals attend to, interpret, and emotionally respond to potentially meaningful events. In line with theoretical frameworks emphasizing biased cognitive processing in insecure attachment, our results suggest that people higher in attachment anxiety and avoidance do not simply perceive hypothetical life events differently, but they perceive and evaluate those circumstances in ways that reinforce their existing relational beliefs. Thus, rather than serving as corrective experiences, life events may function psychologically as confirmatory evidence that maintains—rather than disrupts—attachment orientations [26, 28, 30, 31, 83-85].

Individuals higher in attachment anxiety appeared particularly likely to interpret life events as impactful, emotionally significant, and potentially destabilizing, consistent with hyperactivating strategies that prioritize vigilance, emotional amplification, and threat sensitivity. Their greater belief that events could meaningfully alter personality may reflect an underlying sense of vulnerability and uncertainty about relational stability [17, 47]. Given their tendency to scrutinize relational cues and interpret ambiguous information as threatening or rejection-relevant [30], anxious individuals may perceive life events—especially those involving transition or uncertainty—as opportunities for potential loss, change, or validation of their feared expectations [76]. This aligns with work showing that individuals high in anxiety engage in rumination, catastrophizing, and selective attention toward negative relational cues, which may contribute to the perception that life experiences have powerful and lasting consequences [86]. Taken together, the current findings suggest that anxious individuals’ heightened emotional and cognitive engagement with life events may increase the likelihood that such events shape their relational beliefs over time.

Attachment avoidance was more robustly associated with personality change perceptions. Conversely, though, people higher in attachment avoidance perceived nearly all life events as low in personal relevance and unlikely to influence their personality. This pattern is consistent with deactivating strategies aimed at minimizing reliance on others, suppressing emotional engagement, and maintaining psychological distance [87-89]. Avoidant individuals’ tendency to downplay the relational and emotional significance of events may serve as a defensive mechanism that protects their preferred autonomy and reduces exposure to vulnerability. Prior research demonstrates that avoidant adults often exhibit attentional disengagement, reduced emotional recall, and reinterpretation of relational experiences as inconsequential [44, 54, 55]. All of these behavioral tendencies may contribute to the belief that external circumstances—including potentially disruptive or meaningful ones—pose little threat to their self-concept. In this way, avoidance may buffer individuals from perceived relational instability but, paradoxically, may also prevent them from using life experiences as opportunities for growth or relational recalibration.

Finally, there are a few additional sentences throughout the Discussion that tie the findings back to these recommended sections.

2. Nuancing the Link to Personality (Big Five)

The authors briefly mention the work on Big Five traits but do not fully integrate this into their narrative. This is a missed opportunity. Given the well-established correlations (Attachment Anxiety ~ Neuroticism; Attachment Anxiety and Attachment Avoidance ~ low Agreeableness), the current findings could be partly explained by these broader traits.

For instance, is the heightened negative perception in anxious individuals merely a reflection of their neuroticism? The discussion should explicitly address this potential confound and argue for the unique contribution of attachment working models over and above the Big Five. A control analysis (e.g., partial correlations controlling for Neuroticism) would be a powerful addition, though even a theoretical discussion of this point is necessary.

We acknowledge that links between attachment orientations and Big Five personality traits could have been made more explicit. We have now expanded the section on Big Five personality traits and life event perceptions. We now more deliberately describe the overlap between attachment and personality, including the trait-specific associations the reviewer recommends. However, we also provide some additional discussion about the unique contribution that attachment orientations provide for life and relational outcomes, over-and-above the Big Five personality traits. This addition can be found in lines 138-155 and is reproduced below:

There have also been links established between the Big Five personality traits and these life event perceptions [71]. Reviewing associations with the Big Five personality traits is valuable given their empirical overlap with attachment orientations and because they share some of the same attributional and evaluative tendencies mentioned above [72]. For example, highly agreeable people—a characteristic shared by those lower attachment avoidance—

---

## [Editor Report · Decision Letter 1]

17 Dec 2025

Attachment-related Perceptions of Life Events

PONE-D-25-36404R1

Dear Dr. Chopik,

We’re pleased to inform you that your manuscript has been judged scientifically suitable for publication and will be formally accepted for publication once it meets all outstanding technical requirements.

Kind regards,

Tobias Otterbring

Academic Editor

PLOS ONE

Additional Editor Comments (optional):

Dear authors,

Thank you for delivering a responsive revision. Given the changes implemented in the manuscript as part of this major revision as well as your detailed replies to the reviewers, I am happy to recommend acceptance of your paper in its current form. Congratulations!

Kind regards,

Tobias Otterbring

Associate Editor, PLOS One

---

## [Editor Report · Acceptance letter]

PONE-D-25-36404R1

PLOS One

Dear Dr. Chopik,

I'm pleased to inform you that your manuscript has been deemed suitable for publication in PLOS One. Congratulations! Your manuscript is now being handed over to our production team.

Kind regards,

on behalf of

Professor Tobias Otterbring

Academic Editor

PLOS One